# A shape-driven reentrant jamming transition in confluent monolayers of synthetic cell-mimics

Pragya Arora [1] ✉, Souvik Sadhukhan [2], Saroj Kumar Nandi [2], Dapeng Bi [3], A. K. Sood [4,5] & Rajesh Ganapathy [5,6] ✉

Many critical biological processes, like wound healing, require densely packed cell monolayers/tissues to transition from a jammed solid-like to a fluid-like state. Although numerical studies anticipate changes in the cell shape alone can lead to unjamming, experimental support for this prediction is not definitive because, in living systems, fluidization due to density changes cannot be ruled out. Additionally, a cell's ability to modulate its motility only compounds difficulties since even in assemblies of rigid active particles, changing the nature of self-propulsion has non-trivial effects on the dynamics. Here, we design and assemble a monolayer of synthetic cell-mimics and examine their collective behaviour. By systematically increasing the persistence time of self-propulsion, we discovered a cell shape-driven, density-independent, re-entrant jamming transition. Notably, we observed cell shape and shape variability were mutually constrained in the confluent limit and followed the same universal scaling as that observed in confluent epithelia. Dynamical heterogeneities, however, did not conform to this scaling, with the fast cells showing suppressed shape variability, which our simulations revealed is due to a transient confinement effect of these cells by their slower neighbors. Our experiments unequivocally establish a morphodynamic link, demonstrating that geometric constraints alone can dictate epithelial jamming/unjamming.

Epithelial cell monolayers and tissues are a maximally crowded environment; they are confluent, i.e., the cell packing fraction is almost unity. And yet, remodeling and repair occur, and cancer cells migrate from a tumor and invade distal sites. These processes require tissue/cell collectives to flow locally[1–8], and the jamming-unjamming transition, akin to the one seen in inert particle assemblies[9], provides a pathway. In inert particle assemblies, reducing the density to minimize crowding results in unjamming[10,11], and there is evidence of a similar density-driven transition during embryonic morphogenesis[12] and cancer invasion[13–15]. Besides this conventional unjamming pathway, cells, unlike inert particles, can also deform to surmount the constraining effects of crowding and help the system fluidize[6–8,16]. In fact, in the vertex model of confluent epithelia[17,18], the competing effects of cell contractility and cell-cell adhesion lead to a density-independent but cell shape change-driven unjamming transition[19]. In the jammed state, the cells have a more regular hexagonal shape, whereas in the fluid state, they are more elongated. A subsequent model that included cell motility showed additional factors could drive the jamming

[1]Chemistry and Physics of Materials Unit, Jawaharlal Nehru Centre for Advanced Scientific Research, Jakkur, Bangalore 560064, India. [2]Tata Institute of Fundamental Research, Hyderabad 500046, India. [3]Department of Physics, Northeastern University, Boston, MA 02115, USA. [4]Department of Physics, Indian Institute of Science, Bangalore 560012, India. [5]International Centre for Materials Science, Jawaharlal Nehru Centre for Advanced Scientific Research, Jakkur, Bangalore 560064, India. [6]School of Advanced Materials (SAMat), Jawaharlal Nehru Centre for Advanced Scientific Research, Jakkur, Bangalore 560064, India. ✉e-mail: pragyaarora26@gmail.com; rajeshg@jncasr.ac.in

transition, but the qualitative shape-based nature remained unchanged[20]. Cell shape is proving to be a structural marker of unjamming[6–8,16,21], and this is the case even when alignment interactions between neighboring cells are present[22–25]. Cell shape-mediated unjamming is now implicated in the pathophysiology of asthma[26] and tumor progression[15,27].

Besides their shape, cells in a collective also show substantial shape variability, which, until recently, was dismissed as noise. Atia et al.[28], observed that across vastly different epithelial systems, the cell shape distribution became progressively less skewed as the system jammed. When scaled appropriately, the shape distributions across all the systems collapsed to a $k$-gamma distribution, implying an underlying universality. Such $k$-gamma distributions, interestingly, also arise in the packings of inert materials[29]. Furthermore, and remarkably, across these different epithelial systems, the cell shape variability and

the cell shape followed a simple linear relationship - the larger the cell aspect ratio, the greater the shape variability - that arises for purely geometrical reasons and is insensitive to system details. Taken together, these results suggest that, like in inert particle packings[29,30], geometric constraints take center stage even in tissue jamming/unjamming.

Tissues, however, are complex, and besides cell shape changes, jamming-unjamming due to density changes caused by cell division, apoptosis, extrusion, and cell size changes cannot be ruled out. Further, fluidization can also stem from the presence of motile topological defects[31]. These processes may operate independently or work in cohorts[28] and are difficult, if not impossible, to suppress. Not surprisingly, there is a lack of consensus on whether unjamming is driven by cell shape changes alone[6,8]. Additionally, cells can regulate their motility, which is another critical parameter governing glass/

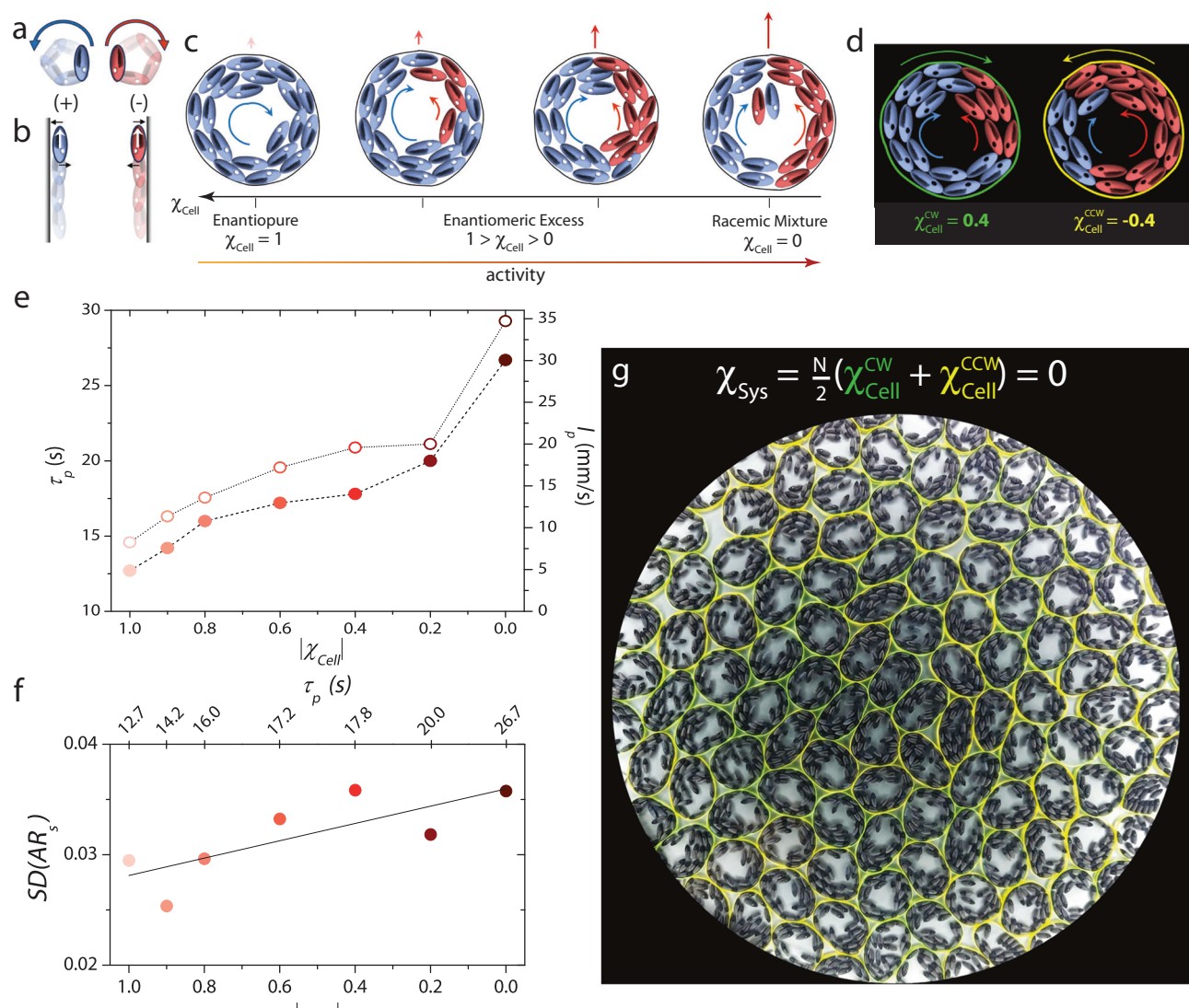

**Fig. 1 | Making a confluent layer of synthetic cell-mimics. a** Superimposed snapshots showing a nearly circular path traced by polar chiral active ellipsoids under vertical agitation. Clockwise (+) and counterclockwise (−) moving ellipsoids are represented in blue and red colours, respectively. **b** Polar chiral active ellipsoids align with their propulsion axis parallel to the wall with the direction of motion along the wall determined by the ellipsoid's handedness. **c** Granular cell enclosing $N = 20$ polar chiral active ellipsoids. Panels from left to right show cells with the chirality of the cell interior, $\chi_{Cell}$, varying from enantiopure to racemic. While a unidirectional polarized wall current results in cell spin, counter-propagating

currents result in cell motility, the extent of which can be tuned by systematically decreasing $|\chi_{Cell}|$. **d** The handedness of cell spin is determined by the sign of $\chi_{cell}$. The green and yellow membranes spin in clockwise and anticlockwise directions due to an excess of (+) ellipsoids in one and (−) ellipsoids in the other. **e** Persistence time $\tau_p$ (hollow circles) and the persistence length $l_p$ (solid circles) versus $|\chi_{cell}|$ of isolated cells. **f** Shape variability of isolated cells, $SD(AR_s)$, versus $|\chi_{cell}|$. The greater the cell activity, the greater the shape variability. **g** Snapshot of a nearly confluent assembly ($\phi \approx 0.92$). The net chirality of the system, $\chi_{Sys} = 0$. The plate has an equal number of clockwise (green) and counterclockwise (yellow) spinning cells.

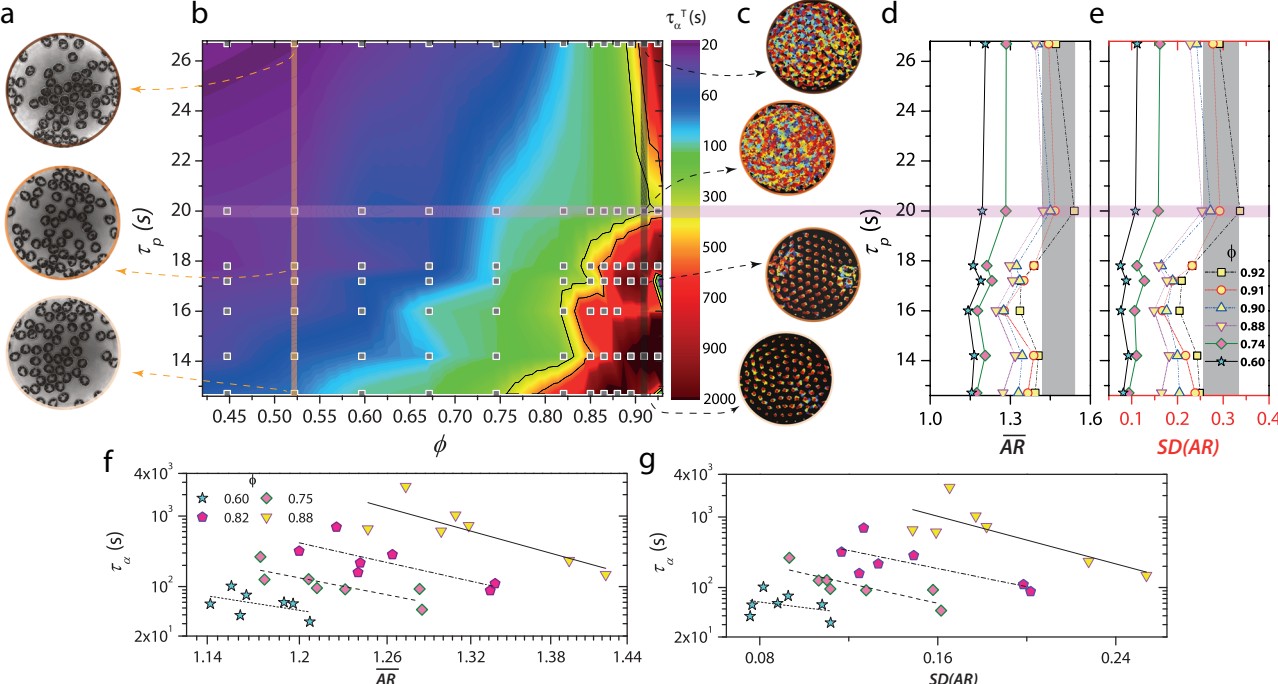

**Fig. 2 | Cell shape and shape variability are correlated with dynamics for a confluent cell monolayer. a** Snapshots of the granular cell assemblies at $\phi = 0.44$ for three representative activities. Note a weak tendency of the cells to form clusters with increasing $\tau_p$. **b** Relaxation dynamics phase diagram in the $(\phi, \tau_p)$ plane. The squares represent the $\chi_{Cell}$ and $\phi$ at which experiments were performed. The black dashed lines are the isochrones. The colour bar indicates the value of the relaxation time $\tau_\alpha$. The color map was constructed from a logarithmic interpolation of the experimentally obtained $\tau_\alpha$ values. We observed a reentrant in dynamics for the four largest values of $\phi$ studied: $\phi = 0.88, 0.90, 0.91, \& 0.92$. **c** Individual cell trajectories in a confluent cell monolayer for $\phi = 0.91$ at different values of $\tau_p$. **d**, **e**, and $SD(AR)$ for different $\tau_p$ values are shown for a range of $\phi$. The window of $\phi$ values where the re-entrant behavior was seen is represented by yellow-filled symbols. Both $\overline{AR}$ and $SD(AR)$ are largest at the intermediate value of $\tau_p$ (horizontal bar), where the system exhibits fluid-like behavior. **(f) & (g)** $\tau_\alpha$ versus $\overline{AR}$, and $\tau_\alpha$ versus $SD(AR)$ for different values of $\phi$.

jamming physics[32,33]. For instance, even for the more straightforward case of rigid self-propelled particles, tuning the activity at constant density results in non-intuitive behavior, such as re-entrant dynamics[34–36]. It is unknown whether similar physics is at play in cell collectives since the means to tune activity systematically are not available here. There is, thus, a clear need for synthetic model systems, wherein parameters like cell deformability[19,20], motility[20,32,33], and number density[32,34], deemed important in theory/numerics, can be controlled precisely. Besides helping bridge the divide between simple models and complex living systems, appropriately designed synthetic systems can potentially single out the role of cell morphology in jamming/unjamming. In fact, a recent theory inspired by[28] posits that the relationship between shape variability and cell shape is purely a mathematical property of a confluent monolayer of closed-loop objects[37]. Still, this idea remains untested due to the lack of a synthetic model system.

## Results

### Making deformable cell-mimics with tunable activity

With these goals in mind, we created synthetic active cell-mimics, assembled them into monolayers, and examined their collective behavior. At the single-cell level, our mimics possess just two features: deformability and tunable activity. Existing deformable active matter systems - centimeter-sized bots confined within paper/metal rings[38] and active colloids confined within vesicles[39] do not serve our purpose. In the former system, each cell is itself large (≈30 cm in diameter), making it a challenge to study collective behavior; with the latter system, methods to create dense assemblies of these and regulate their activity are unavailable.

Our cell-mimics instead are flexible paper rings (3 cm in diameter) that enclose a monolayer of 3D-printed granular ellipsoids rendered chiral active by vertical vibration. These ellipsoids experience both an active torque and force under such driving and perform circle active motion with the handedness of the trajectory, i.e., clockwise (+) or counterclockwise (−), being set when particles are placed manually on the shaker apparatus[40] (Fig. 1a top panel & see Methods and Supplementary Fig. 1–4). Our preliminary observations guided us to leverage chiral activity rather than achiral activity to make our cells self-propelled. On confining achiral polar active ellipsoids within the paper rings, the interplay of membrane curvature and particle orientation often resulted in their accumulation at diametrically opposite ends with their polarity pointing outwards[38,41,42]. Besides resulting in low cell motility, this preferential accumulation led to a nonuniform cell stiffness (Supplementary Fig. 4 and Supplementary Movie 1). In contrast, when persistent active torques are also present, particles hug a wall with their propulsion axis parallel to it and with the direction of motion along the wall set by the handedness of the activity (Fig. 1a bottom panel)[43]. Supplementary Movie 2 shows the dynamics of a granular cell with $N = 20$ chiral ellipsoids of the same handedness, (−), within (Fig. 1c(i)). Unlike the achiral ellipsoids, the particles now uniformly decorate the membrane interior with the polarized particle current acting like a "dynamic internal skeleton". However, this current only caused the paper ring to spin with a well-defined handedness, but we observed little translation.

To tune the motility of the cell-mimics, we changed the handedness of a few ellipsoids inside the membrane from (−) to (+) (red ellipsoids in Fig. 1c(ii)-(iv)). We observed counter-propagating particle currents that suppressed cell spin and made it motile (see Supplementary Movie 2 and Supplementary Table I). Thus, even when $N = N_+ + N_-$ is held fixed at twenty particles, changing the magnitude of chirality of the cell interior, $|\chi_{cell}|$, helps tune cell activity. Here, $\chi_{cell} = \frac{N_+ - N_-}{N_+ + N_-}$, and $N_+$ and $N_-$ are the number of (+) and (−) ellipsoids,

respectively. We note that the handedness of cell spin depends on the sign of $\chi_{cell}$, but the internally generated active force depends only on its magnitude. For example, the green and yellow cells depicted in Fig. 1d spin in opposite directions due to an excess of (−) ellipsoids in one and (+) ellipsoids in the other; however, the active force is determined solely by $|N_+ - N_-|$, which is the same for both.

We quantified the activity of isolated cells for different $|\chi_{cell}|$ values by working in the low area fraction limit of cells, $\phi < 1\%$. We measured the persistence time, $\tau_p$ - the time at which the cells' mean-squared displacement crossed over from ballistic to diffusive dynamics (Supplementary Fig. 5a), and also the average cell speed, $v$ (Supplementary Fig. 5b)[36]. A larger value of $\tau_p$ signifies a stronger departure from equilibrium, i.e., *greater* activity. When the cell interior was gradually changed from enantiopure ($|\chi_{cell}| = 1$, homochiral) to racemic ($|\chi_{cell}| = 0$, 50:50 mixture of (+) and (−) ellipsoids), both $\tau_p$ and the persistence length, $l_p = v\tau_p$, increased in a systematic manner (Fig. 1e, Supplementary Table II). Recent studies have identified $\tau_p$ as a crucial parameter governing active glass physics[33,34,36,44], and our approach allows for systematically tuning it by changing $|\chi_{cell}|$. Interestingly, the active torques and forces exerted by the particles on the membrane interior also influenced the shape variability of isolated cells. We used the standard deviation, $SD$, of an isolated cells' aspect ratio, $AR_s$, as a measure of shape variability, and this is shown in Fig. 1f for different $|\chi_{cell}|$ values. In the limit of an enantiopure cell interior, the persistent torques due to the unidirectional particle current at the boundary suppressed cell shape fluctuations, resulting in small $SD$, while on approaching a racemic cell interior, the more frequent reorganization of the counter-propagating particle currents made it floppy (large $SD$, see Supplementary Movie 2).

## Cell-mimic collective

Before we can ascertain if our synthetic cells in the dense limit embody the crucial features of confluent epithelia, another parameter, namely, the net chirality, $\chi_{Sys}$, of the synthetic cell assemblies must be adjusted. Here, $\chi_{Sys} = \frac{|N_{CW} - N_{CCW}|}{N_{CW} + N_{CCW}}$, where $N_{CW}$ and $N_{CCW}$ are the number of clockwise and counterclockwise spinning cells. We observed that $\chi_{Sys}$ had a qualitative effect on the dynamics; for example, in a nearly confluent assembly ($\phi \approx 0.94$) of clockwise spinning cells, i.e., $\chi_{Sys} = 1$ (green cell in Fig. 1d), there was an emergent edge current (Supplementary Movie 3) like those seen in other confined chiral active matter systems[45,46]. However, except in very rare cases and specific cell lines[47,48], most experiments examining the jamming-unjamming of confluent epithelia do not report such edge currents[1,2,26,28], suggesting that these assemblies do not possess an overall chirality. To make contact with these studies, we, therefore, set $\chi_{Sys} = 0$ by having an equal number of clockwise (green) and counterclockwise (yellow) spinning cells on the plate for all values of $\phi$ and $\tau_p$ studied (see Fig. 1g for $\phi = 0.92$). The edge flows were absent in these assemblies. (Supplementary Movie 4).

## A re-entrant jamming transition mediated by cell shape

Next, by analyzing the dynamical trajectories of the cell centers (see Materials and Methods), we charted the relaxation dynamics of the granular cell assemblies for different values of $\phi$ and $\tau_p$ (Fig. 2a–c). We determined the structural relaxation time, $\tau_\alpha$, as the time at which the self-intermediate scattering function $F_s(q, t)$ decayed to $\frac{1}{e}$ [49] (Supplementary Fig. 6). Here, $q$ is the wavevector and was chosen to correspond to the inverse of the cell diameter. While for a fixed value of $\tau_p$, increasing $\phi$ results in dynamical slowing down, as expected, the effects of increasing $\tau_p$ at fixed $\phi$ is more subtle (Fig. 2b). For $0.4 \leq \phi \leq 0.88$, an increase in $\tau_p$ sped up the dynamics, and over a narrower window $0.4 \leq \phi \leq 0.7$, we observed a weak tendency of the cells to form transient clusters (Fig. 2a and Supplementary Movie 5). This clustering without attractive interactions is a cardinal feature of active matter[50,51] and was evident as a shoulder in the distribution of Voronoi areas (Supplementary Fig. 7). Notably, for the four of the largest densities

studied and which are close to confluence ($0.88 \leq \phi \leq 0.92$), increasing $\tau_p$ resulted in a re-entrant behavior: structural relaxation was fastest at an intermediate value of $\tau_p$ (horizontal bar in Fig. 2b). This re-entrant behavior is evident in the cell trajectories, with the cells being strongly caged (glass-like) for both small and large values of $\tau_p$ and ergodic (fluid-like) for an intermediate value (vertical bar corresponding to $\phi \approx 0.91$ in Fig. 2c, see Supplementary Movie 6).

On increasing the strength of particle attraction in dense assemblies of hard passive particles, the competing effects of particle crowding and attraction often lead to re-entrant dynamics[52,53]. Notably, similar behavior is seen in dense assemblies of hard active particles on increasing $\tau_p$ [32,34–36], with the difference being that the attraction itself is activity-mediated, and its strength is proportional to $\tau_p$. However, unlike in hard passive/active particle systems, where the re-entrant behavior begins to manifest even at moderate densities[34,36,53], we found it only on nearing confluence, which suggested that cell morphology changes may have a vital role here. We calculated the average cell aspect ratio $\overline{AR}$ of the assemblies for a range of $\phi$ and all values of $\tau_p$ studied (Fig. 2d). Remarkably, over the window of $\phi$ values where we observed a re-entrant behavior (yellow-filled symbols in Fig. 2d), it is at the intermediate value of $\tau_p$ where the system was fluid-like that $\overline{AR}$ is largest (horizontal bar in Fig. 2d). Even in our synthetic system, cell shape governs structural relaxation: assemblies of elongated cells relax faster (Fig. 2f)[15,19,20,26]. Indeed, for moderate densities ($\phi = 0.75, 0.60$), not only does the lack of re-entrant dynamics coincide with the absence of a non-monotonicity in $\overline{AR}$ with $\tau_p$ (Fig. 2d), the correspondence between $\tau_\alpha$ and $\overline{AR}$ is also weaker (Fig. 2f).

The above findings suggest that shape variability may also bear the imprint of the dynamics of these assemblies[28]. We quantified the shape variability of our cell assemblies via the standard deviation of the aspect ratio $SD(AR)$. Strikingly, $SD(AR)$ mirrors the behavior of $\overline{AR}$ for different values of $\phi$ and $\tau_p$ (Fig. 2e). Additionally, similar to $\overline{AR}$, there is a correlation between $SD(AR)$ and $\tau_\alpha$, with faster relaxation observed in assemblies with greater shape variability (Fig. 2g).

We can now explain the observed re-entrant behavior. For small $\tau_p$ values (large $|\chi_{cell}|$), the cell motility is small due to the unidirectional particle wall current within the membrane interior (see Fig. 1c). Additionally, due to the very nature of this wall current, each cell is effectively more rigid, i.e., resists significant shape fluctuations (Fig. 1f) and also takes a more disk-like shape, consistent with the observed small value of $\overline{AR}$. Packings of disks jam at smaller densities than of elongated particles since in the former, only the translational degrees of freedom (DOF) need to be frozen out, and this requires fewer constraints than freezing both the translational and orientational DOF as in the latter[8,54]. Therefore, for small $\tau_p$ values, the system is glassy (large $\tau_\alpha$) near confluence. At the largest value of $\tau_p$ (i.e., $|\chi_{cell}| = 0$), due to the frequent reorganization of the particle wall current (see Supplementary Movie 2), individual cell motility is large. The cells are also floppier (Fig. 1f). This larger cell motility results in a strong activity-mediated adhesion[32,34–36] between the cells. Near confluence, due to cell-cell adhesion, the individual cell shape fluctuations are suppressed (Fig. 2 d and e), and the dynamics slow down. The system is again glassy. However, these effects compete for an intermediate $\tau_p$ value: individual cell motility and shape variability are reasonably large, but the cell-cell adhesion is weak, making the assembly fluid-like. As far as we know, this is the first observation of a cell shape-mediated re-entrant jamming.

## Shape and shape variability in confluent cell-mimic monolayers

The strikingly similar behavior of $\overline{AR}$ and $SD(AR)$ for different $\phi$ and $\tau_p$ values (Fig. 2d & e) suggests these two quantities are interdependent. Indeed, for our granular cells, $SD(AR)$ scales linearly with $\overline{AR}$ like that observed in vastly different epithelial systems (Fig. 3a)[28,37]. We checked whether this linear scaling was simply an outcome of the probability distribution function (*PDF*) of the aspect ratio being universal and a $k$

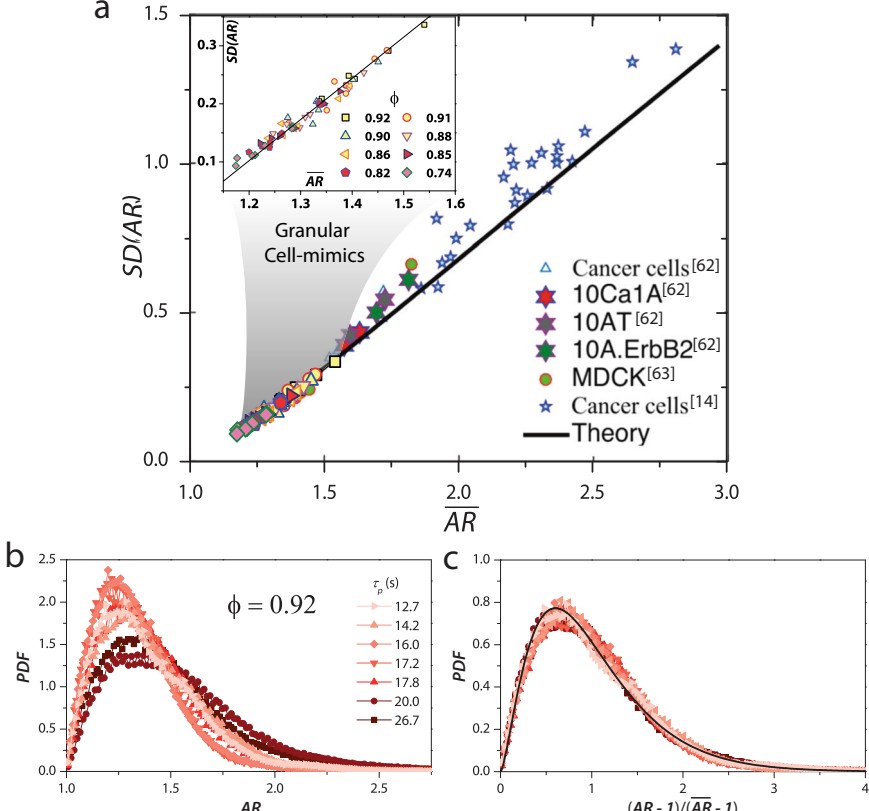

**Fig. 3 | Cell shape and shape variability are mutually constrained for granular cell-mimics. a** $SD(AR)$ versus $\overline{AR}$ for granular cells adhere to the same universal scaling as seen in living systems. Data from our experiments has been overlaid on the figure adopted from ref. [37]; also see[14,73,74]. The black line is the theoretical prediction of ref. 37 and has the functional form ($SD(AR) \simeq 0.71\overline{AR} - 0.75$). **b** Probability density functions (PDFs) of $AR$ for different $\tau_p$ values for $\phi = 0.92$.

Note that *PDF* exhibits a broad distribution with a large positive skew for the unjammed (fluid-like) state at an intermediate value of $\tau_p = 20$ s. For the more jammed states, corresponding to other $\tau_p$ values, the *PDF*s are narrower, and with a smaller skew. **c** When scaled by $(AR - 1)/(\overline{AR} - 1)$, the PDF at different $\tau_p$ values collapse. The black line represents a $k$ − Gamma distribution, and the data can be equally well-fit to the distribution proposed in ref. 37.

−gamma distribution[28,29]. Figure 3b shows the *PDF* for different $\tau_p$ values for $\phi = 0.92$. For the unjammed (fluid-like) state, at the intermediate value of $\tau_p = 20$ s, the *PDF* is broad with a large positive skew. For the more jammed states, corresponding to other $\tau_p$ values, the *PDF*s are narrower and have a smaller skew. This behavior is also seen for other values of $\phi$ near confluence (Supplementary Figs. 8–11). We also ensured that our systems' confining boundary did not have a significant bearing on these results (Supplementary Fig. 12). Moreover, the *PDF*s at different $\tau_p$ values collapse onto a single universal distribution on rescaling $AR$ to $x = \frac{(AR-1)}{(\overline{AR}-1)}$, which ensures that the scaled *PDF*s start at the origin and have a mean of unity (Fig. 3c). Within experimental uncertainty, the scaled *PDF*s are equally well-fit by the $k$−gamma distribution (1) with the value of $k \approx 2.51$[28], as well as the recently posited distribution for a confluent assembly of closed-loop objects (2), which is only *nearly* universal[37] (Supplementary Fig. 13).

$$P(x,k) = \left[ k^k / \Gamma(k) \right] x^{k-1} \exp[-kx] \qquad (1)$$

$$P(AR) = \frac{1}{\mathcal{N}} \left( AR + \frac{1}{AR} \right)^{3/2} \left( 1 - \frac{1}{AR^2} \right) e^{-\alpha \left( AR + \frac{1}{AR} \right)}, \qquad (2)$$

In the first expression, $\Gamma(k)$, is the Legendre gamma function; in the second, $\mathcal{N}$, is the normalization constant, and $\alpha$ is a system-specific parameter. For all values of $\phi$ close to confluence and all $\tau_p$ values, $k$ hovers between 2.5−2.8 like that observed in confluent epithelia[28].

Equation (2) follows from a mean-field theory[37] for a confluent monolayer of closed-loop objects, and as such, it applies to confluent epithelia[37] as well as our synthetic cell mimic collective (Supplementary Fig. 13) Importantly, Eq. (2) depends only on a single parameter, $\alpha$, and this has two immediate consequences: *(1)* The PDF of $x$ obtained from Eq. (2) becomes nearly universal[37] and is consistent with the collapse of the PDFs observed in Fig. 3c. *(2)* Both $SD(AR)$ and $\overline{AR}$ must be functions of $\alpha$ alone. Using these two functions and eliminating $\alpha$ results in a parametric equation, $SD(AR) = 0.71\overline{AR} - 0.75$, that is independent of system details (black line in Fig. 3a)[37]. Indeed, our experiments are in excellent agreement with the theoretically predicted line.

The validity of these predictions in our experiments has several implications. First, one of the central assumptions of the analytical calculations of ref. 37 is that the constraint of confluency is not crucial for Eq. (2); all that is needed is sufficient fluctuations of the cell boundary that permits an effective equilibrium description. In fact, we see excellent agreement between experiments and theory even for densities far from confluence ($\phi = 0.74$) (inset to Fig. 3a) directly verifying this assumption. This agreement, however, becomes weaker for smaller $\phi$ values. Second, systems with slower dynamics (larger $\tau_\alpha$) have smaller values of $SD(AR)$ and $\overline{AR}$. Importantly, as was found in ref. 37, we observed that $\alpha$ scales linearly with $\log(\tau_\alpha)$, indicating a strong link between structure and dynamics (Supplementary Fig. 14).

**Dynamical heterogeneities, cell shape and shape variability**

It is widely recognized that the dynamics of dense passive/active liquids and glasses are spatiotemporally heterogeneous and consist of domains with different relaxation times[34,36,55,56]. However, the correlation between cell aspect ratio and dynamics uncovered here and in previous studies[19,26,28], was from a system-wide averaging. In fact,

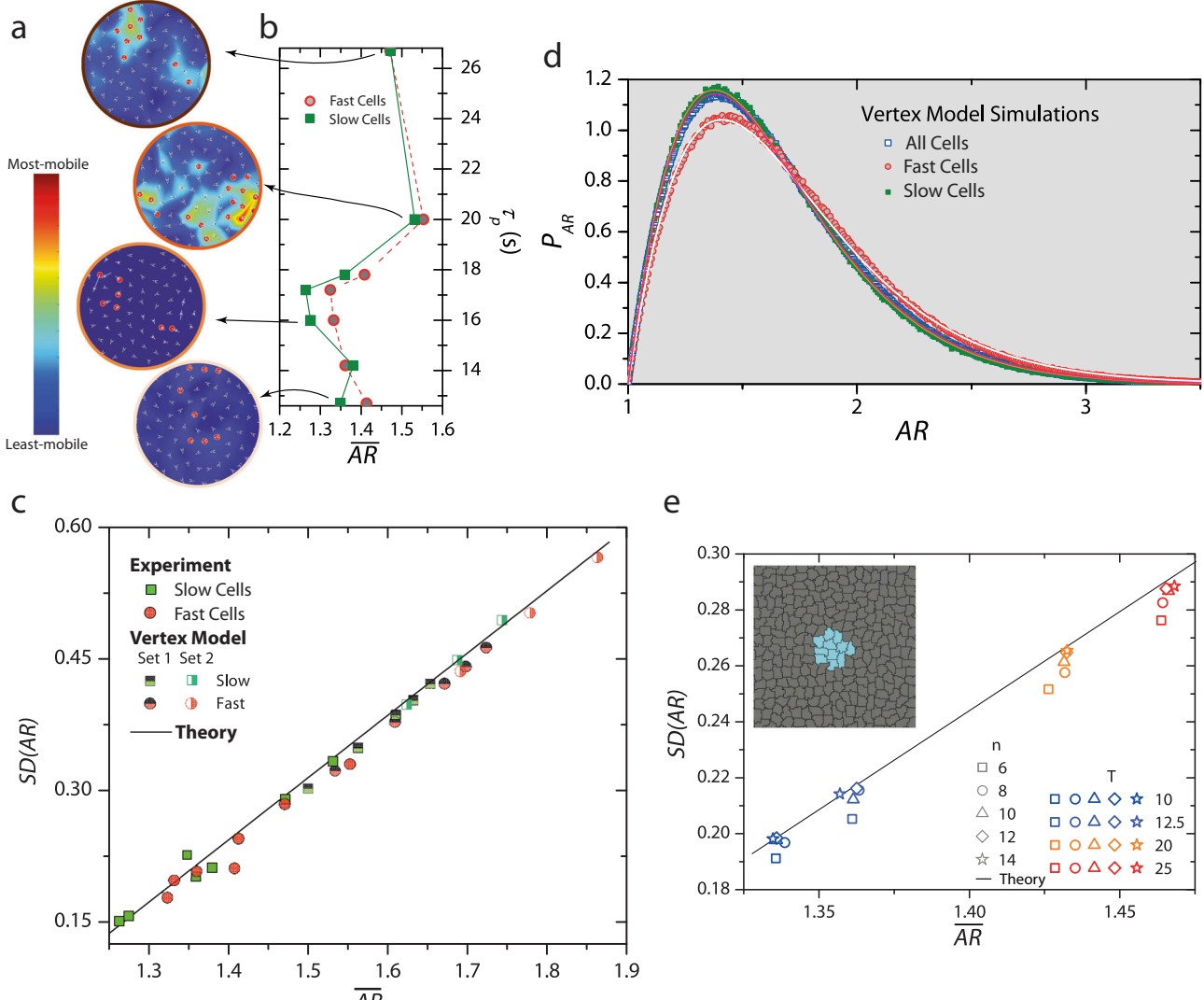

**Fig. 4 | Fast cells show suppressed shape variability. a** Cell velocity maps over the cage breaking time, $t^*$, for different $\tau_p$ values for $\phi = 0.92$. The color bar denotes the displacement magnitude. The red circles represent the top 10% most-mobile cells over $t^*$. **b** $\overline{AR}$ for different values of $\tau_p$ for the top 10% most-mobile (hollow circles) and least-mobile cells (squares) cells over $t^*$. Fast cells typically have a larger $\overline{AR}$ than the slow cells. **c** $SD(AR)$ versus $\overline{AR}$ for the fast (circles) and slow cells (squares). The fast cells show a reduced shape variability in both experiments (filled symbols) and Vertex Model simulations (half-filled symbols). **d** PDF of $AR$ for all cells (hollow squares), 10% fast (red circles) and 10% slow cells (solid squares) from Vertex model

simulations at $T = 0.009$. **e** Inset: Mimicking dynamical heterogeneities. After equilibrating the system, a cluster of $n$ cells (shown in cyan) is allowed to evolve while the rest of the system is frozen (gray). Main figure: $SD(AR)$ versus $\overline{AR}$ for different $n$ and at different $T$. The black line is the universal scaling predicted in ref. 37. With increasing $n$, $SD(AR)$ approaches the universal scaling from below, demonstrating that the subdued shape variability of fast cells is due to their confinement by their slower surrounding. Also, at low $T$, where the mobility of the system as a whole is smaller, the deviation from the line is also small. At high $T$, on the other hand, the confinement effect is more pronounced.

recent experiments on confluent cell monolayers and 3D tissues show that such a correlation is also present locally, i.e., cells in the faster relaxing regions are more elongated than those in the slower relaxing ones[8,15]. Additionally, since at the system level, $SD(AR)$ and $\overline{AR}$ are mutually constrained (Fig. 3a), it is reasonable to expect that cell shape variability in these faster regions is also correspondingly larger, although this remains untested.

To this end, we followed standard procedures to quantify dynamical heterogeneities[55,56] and first identified the top 10% fastest and slowest cells over the cage-rearrangement time, $t^*$ (Supplementary Fig. 15). In Fig. 4a, we show the fastest cells (red circles) overlaid on the cell displacement maps also generated over $t^*$ for different $\tau_p$ values for $\phi = 0.92$. As is typical of deeply supercooled liquids, the fast and slow cells formed separate clusters, and further, these fast cells within a cluster move together largely as a flock. Also, the heterogeneity was most pronounced at an intermediate value of $\tau_p$ (Supplementary

Movie S7). Next, we determined the mean and standard deviation of the aspect ratio of the cells within these fast and slow clusters. Notably, fast cells have a larger $\overline{AR}$ than slow cells (Fig. 4b). However, while the growth of $SD(AR)$ with $\overline{AR}$ for the slow cells followed the universal scaling, for the fast cells, surprisingly, it lies below the predicted line, implying that these cells show reduced shape variability (solid symbols in Fig. 4c).

To strengthen these findings and examine their generality, we performed simulations of the Vertex model (VM)[18] and the Cellular Potts model (CPM)[57], which are highly effective representations of confluent epithelia. Simulation parameters were chosen so that the relaxation times of the systems were comparable to those in the experiment. The larger system size and superior temporal statistics in our simulations also allowed quantifying the *PDF* for the fast and slow cells. We observed that while the *PDF* of the slow cells overlapped with that of the entire system, the *PDF* for the fast cells had a larger skew, as

is expected for a locally unjammed region of the system (Fig. 4d and Supplementary Fig. 16). It is noteworthy that even in the simulations, and for different sets of parameter values, the growth of $SD(AR)$ with $\overline{AR}$ for slow cells followed the predicted scaling, while for fast cells, $SD(AR)$ was lower than anticipated (half-filled symbols in Fig. 4c and Supplementary Fig. 17). A linear regression-based error analysis lends further support to these observations (Supplementary Section XI and Supplementary Fig. 18).

The fact that the most-mobile cells do not conform to this universal scaling is likely because it is from a theory that is mean-field in character - all cells are considered equivalent, and spatio-temporal heterogenities in cell mobilities are ignored. Mean-field approaches fail to capture the physics of dynamical heterogeneities even in conventional dense liquids and glasses[55]. Nonetheless, to gain insights into our observations, we made the simplifying assumption that on a time scale comparable to $t^*$, fast cells are essentially confined within a frozen exterior of slower cells due to their different relaxation rates. To mimic this effect numerically, we first allowed the entire system to equilibrate over many times $\tau_\alpha$, and then froze all but a cluster of $n$ cells in the system interior (shown in cyan in the inset to Fig. 4e). We performed simulations at different temperatures to let the system access different values of $\overline{AR}$. Trivially, when all but one cell is frozen ($n = 1$), there can be no shape variability, and $SD(AR) = 0$. Indeed, on increasing $n$, $SD(AR)$ increased systematically and followed the predicted scaling for $n \geq 12$. This observation nicely demonstrates that the subdued shape variability of the fast cells is an outcome of the temporary confinement imposed by their slower neighbors.

## Discussion

Our confluent synthetic cell monolayers are an oversimplification of confluent epithelia. Unlike in the latter, no explicit cell-cell adhesion exists, although an effective activity-mediated attraction emerges here. These synthetic cells also have a fixed perimeter, which is not the case with live cells. The area constraint in our system is weaker than in confluent epithelia but still exists because of a dynamic internal skeleton. Nonetheless, in the context of jamming-unjamming, our synthetic cell collective captures some crucial features of living ones[15,26–28]. The most striking is that at the system level, cell shape variability scales with the average aspect ratio in a manner identical to that observed in confluent epithelia[28] and in line with mean-field predictions[37]. Dynamical heterogeneities, however, violate these predictions, with the sub-population of fast cells showing subdued shape variability. To fully describe even a minimal model system like ours, there is a clear need to go beyond mean-field approaches. Importantly, while our observation of a reentrant jamming transition with increasing persistence time is reminiscent of those seen in dense assemblies of hard active particles, here, it is mediated purely by cell shape changes. Taken together, these results imply a morphodynamic link underlying jamming-unjamming.

On the experimental front, the system introduced here allows precise control over many critical physical parameters like the chirality of the assembly, individual cell membrane stiffness, the nature of activity within the membranes, and even the inter-cell friction. Although each of these parameters has been identified to substantially impact the form and function of the cell collective[58–60], these are impossible to control systematically in real systems, and some of these, like inter-cell friction, are yet to be incorporated even within the models of confluent epithelia. We can already modify individual cell properties in real-time in our preliminary experiments with magnetic membranes. The experimental advance made here now makes it possible to probe the role of heterogeneous cell properties on collective behavior[61] and in fundamental processes like cell sorting[62] from a purely geometric/mechanical perspective[63–65].

## Methods

### Making deformable cell-mimics
Our deformable cell-mimics were made by confining millimeter-sized 3D-printed granular ellipsoids within flexible paper rings. The paper rings, measuring 3.2 cm in diameter and with a thickness of 40−50 $\mu m$ (Supplementary Fig. 1), were made by gluing the ends of thin paper strips, each 10 cm in length and 0.21 cm in height. The granular ellipsoids confined within these paper rings were 3D printed using the PROJET 3600 Multijet 3D printer. This printer employs an inkjet printing process utilizing piezo print-head technology for the sequential deposition of a photocurable plastic resin and casting wax material layer by layer around the particle. The particles were designed using MATLAB and AutoCAD software. The printer operated at a layer resolution of 16 $\mu m$, a scale much finer than the dimensions of the particles, which were in the millimeter-range, thereby enabling precise control over surface characteristics and the realization of intricate structural features. Roughly three thousand ellipsoids can be printed in three hours and in a single batch.

### Designing chiral active granular ellipsoids
Earlier studies found that granules with an asymmetry in shape or mass $m$, friction coefficient, $\mu$, or some combination of these between the two ends of granules, exhibit self-propulsion along the direction set by the asymmetry (Supplementary Fig. 2)[66–71]. Building on these studies here, we employed 3D printing to create plastic ellipsoids with fore-aft asymmetry in $\mu$ and $m$, respectively.

We tested many designs of polar and apolar ellipsoids, and the final design was chosen based on the observed nature of active dynamics. The first step involved introducing friction anisotropy, which was achieved by making one end of the ellipsoid rougher than the other using a feature inherent to the printing process. During the printing process, half of the ellipsoid is embedded within the wax matrix, which serves as support during the layer-by-layer printing (depicted in Supplementary Fig. 3). The wax-covered surface of the ellipsoid is noticeably rougher compared to the exposed portion, thereby introducing friction anisotropy through variations in surface texture. This friction anisotropy resulted in the ellipsoids exhibiting polar activity, with the smoother end (transparent portion) as the head and the rougher end (white portion) as the tail (Supplementary Fig. 2). These particles have a fore-aft asymmetry and are intrinsically polar active along the direction set by the asymmetry.

Additionally, to introduce an asymmetry in $m$ besides the friction asymmetry, a hole was incorporated along the major axis towards the rough end of the particle (i.e., the tail or trailing end). We observed that these ellipsoids were self-propelled along the major axis with the hole at the trailing end. Consequently, the presence of asymmetry in both $m$ and $\mu$ made the particles polar active.

Finally, to impart chiral activity to the particle, it was essential to break the left-right symmetry along the propulsion direction. This was achieved by making one portion of the ellipsoid hollow during print (shown by the dashed red line in Supplementary Fig. 2C).

### Post-processing and cleaning particles
The post-processing procedure for the particles embedded within a wax mold involved subjecting them to sonication in oil at 60 °C, effectively facilitating the removal of the wax matrix. Subsequently, the plastic parts were thoroughly cleaned using a soap solution. The final stage entailed washing the particles with isopropanol, followed by drying.

### Details of the shaker apparatus
The 3D-printed granules were placed on a horizontally mounted aluminum plate connected to an electromagnetic shaker via an air bearing and confined from above by a glass plate to facilitate imaging[72]. The gap between the top and bottom plate, $\Delta$, satisfies $\delta < \Delta < \beta$, which

prevents the particle from flipping. The drive frequency, $f = 37$ Hz, and amplitude, $a = 1$ mm, were maintained constant, resulting in a non-dimensional acceleration of $\Gamma = \frac{4\pi^2 f^2 a}{g} = 5.5$, where $g$ is the gravitational acceleration.

## Imaging and feature-finding

The granular cells are illuminated from above with a custom-made LED light ring. Our arrangement ensures that the lighting is uniform. The data was recorded at frame rates ranging from 0.5-10 Hz, depending on the area fraction of the cells on the plate, and at a spatial resolution of 2464 X 2056 pixels using a CMOS camera (Victorem, IO Industries Canada). We used ImageJ and custom-written codes in Matlab to track the morphology and dynamics of the cells.

## Reporting summary

Further information on research design is available in the Nature Portfolio Reporting Summary linked to this article.

## Data availability

All study data are included in the article or supplementary information. The raw data files, which are in excess of 3 TB, are available from the corresponding authors upon request. The source data files for the main figures generated in this study have been deposited in the Figshare.com database under accession code 10.6084/m9.figshare.25830829.

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

## Acknowledgements

P.A. thanks the Jawaharlal Nehru Centre for Advanced Scientific Research, Bangalore, INDIA, for a research fellowship. S.S. and S.K.N. acknowledge the support of the Department of Atomic Energy, Government of India, under Project Identification No. RTI 4007. SKN thanks SERB for the grant via SRG/2021/002014. D.B. was supported by the US National Science Foundation (DMR-2046683) and the Alfred P. Sloan Foundation. A.K.S. thanks the Science and Engineering Research Board, Government of India for the National Science Chair. R.G. thanks the Department of Science and Technology, INDIA, for financial support through the Swarna Jayanti Fellowship Grant (DST/SJF/PSA-03/2017-22).

## Author contributions

P.A.: Conceptualization, Methodology (experiments), Software, Validation, Formal Analysis, Investigation, Data Curation, Visualization, Writing - Original Draft, Writing - Review & Editing. S.S.: Methodology (simulations), Software, Validation, Formal Analysis, Data Curation, Writing - Review & Editing. S.K.N.: Methodology (simulations), Software, Validation, Data Curation, Formal Analysis, Writing - Review & Editing, Supervision (simulations), Funding Acquisition. D.B.: Formal Analysis, Validation, Writing - Review & Editing. A.K.S.: Validation, Writing - Review & Editing. R.G.: Conceptualization, Methodology (experiments), Validation, Investigation, Formal Analysis, Visualization, Writing - Original Draft, Writing - Review & Editing, Supervision (experiments), Project Administration, Funding Acquisition.

## Competing interests

The authors declare no competing interests.

## Additional information

**Article**

