## [Peer Review File · Nature Communications]

REVIEWER COMMENTS

Reviewer #1 (Remarks to the Author):

This paper reports results on how flexible mobile particle monolayers behave. Both the persistence time of the mobility of the considered entities as well as their 2D surface fraction have been changed. The interesting and new result is the observed fluidity of the monolayer at intermediate values of the persistence time and relatively high surface fractions. What is even more interesting is the reentrant nature of this fluidity as the persistence time is changed.

The study is novel in many ways, both in the results obtained as well as the design of the system used. Even if it is inspired by recently proposed systems using flexible walls and membranes enclosing active macroscopic particles or microscopic colloids, the system devised here has its own interest and properties and allows for a systematic investigation of the interplay between deformability, mobility and density.

Overall this study seems robust and will interest diverse communities in both active matter and possibly biophysics.

I am willing to reconsider this paper once the authors consider the following recommendations and comments.

1. The main plot of the paper (Fig.2b) has little squares indicating the measurement points carried out by the authors. The color map of relation time is apparently based on these measurements using some linear interpolation scheme. I am not sure that I fully understand how this color map was constructed and simple linear interpolation between the scarce points on this 2D plot does not seem plausible. Can the authors explain this in detail?
2. In the main text, there is a discussion of some mean field theory relevant to the results of the paper including the relation between AR and $S_d(AR)$. The text is silent as to what the significance of this is and what such relations tell us about the system considered. Since this is apparently an important aspect of the results obtained, the authors should explain what this theory is all about.
3. The discussion about dynamical heterogeneities is certainly of interest but as it is now it is not clear whether the observed effects are significant (Fig.4c) and the use of numerics even though of possible interest is not fully justified. In any case Fig. 4 is not very convincing as it is now. The authors need to strengthen this section if the claims made are to be maintained.

On a minor note, the authors mention some t^* in the main text but I am not sure I found how it is defined, also the values of τ_p are given but I am not sure I found how this time scale is obtained from the experiments.

Reviewer #2 (Remarks to the Author):

The authors report on a novel experimental approach to study confluent tissues by means of synthetic active particles embedded in a paper strip.

The idea is brilliant and I find it very interesting and useful also for future studies.

The authors have shown that indeed by using their experimental model they can vary different parameters and hence produce diverse dynamical patterns.

However, I do not find this manuscript suitable for publication for the following reasons

1- After having read the manuscript multiple times it is not clear to me what is the main point of the authors. Is the main point of this manuscript to present their experimental set up as a new mean to simulate confluent tissues? Or is it to study certain features of confluent tissues?

The confusion is originated by the mismatch between the generic introduction and the data presented. I suggest the authors to clarify the real goal of this manuscript and to design the text accordingly.

2- The authors make a big publicity to the re-entrant feature that they observe in the phase-space. Is this so important? It seems that the full story and the full study of confluent tissues is about finding re-entrant phases. Moreover: if it is so important., the authors should have said why it is so and they should explain why it occurs in their model. Their current "explanation" (last paragraph of pag.7) is quite vague and non-understandable. All in all which are the physical mechanisms responsible for the re-entrant behaviour is not spelled out clearly

3- In the introduction the authors say "Our experiments unequivocally establish a morphodynamic link, demonstrating that geometric constraints alone can dictate epithelial jamming/unjamming". Where do we see this? If it is so important why this is not commented in the conclusions?

4- The text is not well written:

4.a the authors use very complex syntactic structures and very long sentences that make the reading very complicated. For example (but this is not the only one), I cannot understand the following sentences:

- Even for the simpler case of rigid self-propelled particles, tuning the activity at constant density results in non-intuitive behavior, such as re-entrant dynamics [34–36], and it is unknown whether similar physics is at play in cell collectives, since, here, the means to tune activity systematically are not available. Synthetic models that embody the key features of confluent epithelia can help bridge the divide between theory/numerical predictions and experiments on living systems.

I mean: "similar" to what? which "divide"?

4.b the authors use many name and nomenclature without introducing them.

For example: confluent tissue, enantiopure and racemic are used without introducing them.

5- from time to time it seems that the authors are interested more in promoting their experimental setup rather than the results that they have achieved with it. For example authors say " Recent studies have identified τ_p as a crucial parameter governing active glass physics [33, 34, 36, 44], and our approach allows for systematically tuning it." But then they do not do it..

6- The reentrant behavior that seems to be the most important result is indeed very weak.... the authors should quantify the errorbars of their measurements. Is it a matter of statistics?

How does this scale upon increasing the sample size? is it a finite-size effect? Will it persist for bigger tissues??

All in all after reading the manuscript multiple time I was left with the feeling that the authors stepped on something indeed interesting but they are not putting their results in a way that the reader can appreciate them.

Reviewer #3 (Remarks to the Author):

The submitted manuscript collective dynamics in a monolayer of synthetic cell-mimics which are driven by vibrating granular particles confined in an elastic boundary. The authors systematically increased the persistence time of self-propulsion and discovered a cell shape-driven, density-independent, reentrant jamming transition. They also observed that cell shape and shape variability are strongly related in the confluent limit and followed the same universal scaling as that observed in confluent epithelia. Furthermore, the fast cells show suppressed shape variability and did not conform to this scaling. These results suggest that geometric constraints alone can dictate epithelial jamming/unjamming.

In my opinion, the submitted manuscript contains novel results and is well written. I recommend its publication after the authors consider the following questions.

1. The experiments are fairly limited in size: the radius of the imaging area is about 6-7 cell diameter. In such a small system, we should expect strong boundary effects and a spatial gradient over the system. Indeed, cells in the center region of Fig. 1g appear to be more elongated. This spatial inhomogeneity over the whole system, if exists, makes interpretation of the experimental results more complicated. The authors should discuss how strong confinement influences their results and whether it is proper to compare their results with those from an infinite system.
2. The authors used Potts and Vertex models to understand the role of fast cells. I wonder how the parameters of these models were chosen. Are there any experimental supports for the chosen parameters?
3. The authors reported a re-entrant jamming transition mediated by cell shape. Is it possible to reproduce such a phenomenon in their numerical models?

Comments of Reviewer 1 along with our responses – NCOMMS-24-01719-T/Arora

This paper reports results on how flexible mobile particle monolayers behave. Both the persistence time of the mobility of the considered entities as well as their 2D surface fraction have been changed. The interesting and new result is the observed fluidity of the monolayer at intermediate values of the persistence time and relatively high surface fractions. What is even more interesting is the reentrant nature of this fluidity as the persistence time is changed. The study is novel in many ways, both in the results obtained as well as the design of the system used. Even if it is inspired by recently proposed systems using flexible walls and membranes enclosing active macroscopic particles or microscopic colloids, the system devised here has its own interest and properties and allows for a systematic investigation of the interplay between deformability, mobility and density. Overall, this study seems robust and will interest diverse communities in both active matter and possibly biophysics. I am willing to reconsider this paper once the authors consider the following recommendations and comments.

Response: We thank the referee for finding both the experimental system developed here and the results obtained novel and being willing to reconsider a revised version of the study. Below, we provide a point-by-point response to their comments. All the changes made to the text in the main manuscript and supplement are colored magenta.

Specific comments:

Comment 1: *The main plot of the paper (Fig.2b) has little squares indicating the measurement points carried out by the authors. The color map of relaxation time is apparently based on these measurements using some linear interpolation scheme. I am not sure that I fully understand how this color map was constructed and simple linear interpolation between the scarce points on this 2D plot does not seem plausible. Can the authors explain this in detail?*

Response 1: We apologize for the typo in the caption of Fig. 2 of the Main Manuscript, where we have mentioned a linear interpolation scheme. The referee is correct that the color map cannot be constructed from linear interpolation of the experimental data. Since over the range of area fractions ϕ explored, the translation relaxation time, τ_α , grows by over two orders of magnitude, we have used a color map that followed a logarithmic interpolation. This color map was generated using the standard data plotting software Origin Pro 8.5. The caption to Fig. 2b has now been corrected.

Comment 2: *In the main text, there is a discussion of some mean field theory relevant to the results of the paper including the relation between AR and Sd(AR). The text is silent as to what the significance of this is and what such relations tell us about the system considered. Since this is apparently an important aspect of the results obtained, the authors should explain what this theory is all about.*

Response 2: We thank the referee for this suggestion. We have now expanded this

section to highlight the significance of the findings. Please see *lines 238-256 and 292-294 of the revised main manuscript and Section VIII of the revised supplement*. The mean-field theory Ref. [37] of the main manuscript was inspired by findings in Ref. [28], where it was found that vastly different epithelial systems showed the same scaling between $SD(AR)$ and \overline{AR} . The authors of Ref. [37] arrived at an expression for the PDF of shape variability, $P(AR)$ (Eq. (2) of the main manuscript), and from here derive a universal scaling between $SD(AR)$ and \overline{AR} . Importantly, Ref. [37] posits that this scaling is simply a feature of a confluent layer of closed-loop objects and, as such, applies to both real and synthetic cells. We have indeed shown that it applies to our system as well. Also, for this theory to work, confluency constraint is not crucial, and we indeed find experiments for $\phi = 0.75$ also follow the predicted scaling (inset to Fig. 3a). Furthermore, Eq. (2) depends on a single system-specific parameter, α , that is predicted to correlate with the dynamics. We have tested this claim as well (Supplementary Fig. S14). However, being a mean-field theory, all cells are treated on the same footing, and it does not account for dynamical heterogeneities. We indeed find that the subpopulation of fast cells violates these predictions.

Comment 3: The discussion about dynamical heterogeneities is certainly of interest but as it is now it is not clear whether the observed effects are significant (Fig.4c) and the use of numerics even though of possible interest is not fully justified. In any case Fig. 4 is not very convincing as it is now. The authors need to strengthen this section if the claims made are to be maintained. On a minor note, the authors mention some t^ in the main text but I am not sure I found how it is defined, also the values of τ_p are given but I am not sure I found how this time scale is obtained from the experiments.*

Response 3: We agree with the referee that the observed differences in the scaling of $SD(AR)$ versus \overline{AR} for the fast and slow cells is small. But it is very much present. When we observed this effect, first in our experiments ($\phi = 0.92$), we harbored the same concerns as the referee. To ascertain its veracity, we performed simulations using two well-studied numerical models of confluent epithelia, namely the cellular Potts model and the Vertex Model. In both these models, we again observed that the slow cells adhered to the scaling while the fast ones deviated from it (Fig. 4c and Fig. S16). Simulations for different sets of parameters (mentioned as Set 1 and Set 2 in Fig. 4c) also showed similar results. We apologize for not highlighting this fact in our first submission, which has now been rectified (caption to Fig. 4c). Only after observing this agreement between experiments and simulations did we attempt to rationalize the difference in the behavior of fast and slow cells.

Nonetheless, to further strengthen this section as suggested by the referee, we have now carried out the analysis for another area fraction studied in the experiments ($\phi = 0.90$). We have also analyzed the data for another set of data from CPM simulations. All the analyses from experiments and simulations are now summarized in a *new Supplementary Fig. S18 and Section XI of the revised supplement*. Supplementary Fig. S18 shows the sum of squared errors defined as,

$$SSE_{F,S} = \sum_{i=1}^N (SD(AR)_{\text{Fit}}^{F,S} - SD(AR)_{\text{Th}})^2$$

between the best-fit line to the data for fast (F) and slow (S) cells and the theoretical

prediction of Ref. [37] of the Main manuscript ($SD(AR)_{\text{Th}} = 0.71\overline{AR} - 0.75$). Here, N is the total number of data points in each of the experimental/simulation data sets. We observed that in all sets of data, $SSE_{\text{F}} > SSE_{\text{S}}$, and in some instances by almost an order of magnitude. This analysis unambiguously reveals that fast cells veer off mean-field theoretical predictions and strengthen our observations.

We would also like to emphasize here that this subtle effect, which we are able to pick up, is large because of the accuracy with which we can track the cell perimeter and shape of the synthetic cells. With real cell data, these are often quite challenging.

Regarding the other minor comments raised by the referee, we apologize for any lack of clarity in defining how t^* was obtained from the experiments in the main manuscript. The translational non-Gaussian parameter, $\alpha_2^T(t)$, was used to characterize the heterogeneous dynamics. The time corresponding to the maximum in $\alpha_2^T(t)$ is the cage breaking time t^* , where the dynamics is maximally heterogeneous. We want to highlight that the definition of t^* was a part of the supplement in Section VIII “Quantifying dynamical heterogeneity,” and is now **Section IX of the revised supplementary material**. Supplementary Fig. S15 is cited in line 269 of the main manuscript.

While our main manuscript did mention that the persistence time τ_p was identified with the crossover time from ballistic to diffusive dynamics of the translational mean-squared displacement measured in the dilute limit, the associated supplementary figure lacked the necessary details. We have now rectified this with a revised figure - **Supplementary Fig. S5A** - that clearly shows how τ_p was determined as the single-cell chirality was tuned.

We thank the referee for their constructive comments, which has helped strengthen our findings and improve the readability of the manuscript. We hope that the referee finds our revised manuscript suitable for publication in Nature Communications.

Comments of Referee 2 along with our responses – NCOMMS-24-01719-T/Arora

The authors report on a novel experimental approach to study confluent tissues by means of synthetic active particles embedded in a paper strip. The idea is brilliant and I find it very interesting and useful also for future studies. The authors have shown that indeed by using their experimental model they can vary different parameters and hence produce diverse dynamical patterns. However, I do not find this manuscript suitable for publication for the following reasons

Response: We were very pleased that the referee found our approach to studying some aspects of confluent epithelia “*novel*”, “*brilliant*”, “*very interesting*” and “*useful*”. We have now attempted to fully address the referee’s concerns and a point-by-point response to their comments follows. All changes made to the main text and supplement are colored magenta.

Specific comments:

Comment 1: After having read the manuscript multiple times it is not clear to me what is the main point of the authors. Is the main point of this manuscript to present their experimental set up as a new mean to simulate confluent tissues? Or is it to study certain features of confluent tissues? The confusion is originated by the mismatch between the generic introduction and the data presented. I suggest the authors to clarify the real goal of this manuscript and to deign the text accordingly.

Response 1: We thank the referee for this comment. Our study presents the first synthetic experimental system that can capture some crucial features observed in the confluent epithelia, which are more complex. With our model system, we find features (Fig. 3) that have been reported for confluent epithelia Ref. [28] of main manuscript, as well as new features which include the observation of reentrant dynamics with increasing τ_p and the violation of mean-field scaling by the fast cell population (Fig. 4). Since all our observations pertain to jamming-unjamming and glassy dynamics, our original introduction also pertained to these aspects. Following this comment by the referee, we have kept the essence of the introduction the same but have now made substantial text changes to introduce the problem better and also emphasize the need for a synthetic system (*lines: 42-46, 59-61, 64-65, 68-72, and 77-82 of main manuscript*). Also, we have revamped the conclusions to better convey our results (*lines: 312-322 of main manuscript*). If the referee requires further changes, we will be happy to incorporate these as well.

Comment 2: The authors make a big publicity to the re-entrant feature that they observe in the phase-space. Is this so important? It seems that the full story and the full study of confluent tissues is about finding re-entrant phases. Moreover: if it is so important., the authors should have said why it is so and they should explain why it occurs in their model. Their current "explanation" (last paragraph of pag.7) is quite vague and non-understandable. All in all which are the physical mechanisms responsible for the re-entrant behaviour is not spelled out clearly.

Response 2: We thank the referee for this comment. One of the most striking results that was first reported in experiments and simulations of hard active particles was the observation of clustering without explicit attractive interactions (Ref. [49, 50]). In these systems, clustering is due to the finite persistence time, and subsequent studies reported a reentrant in the dynamics with increasing persistence time (Ref. [32, 34-36]). Up until then, such a type of reentrant dynamics was seen only in passive systems with short-range attractive interactions (Ref. [51, 52]). We report the first observation of reentrant dynamics in a deformable active matter system on increasing the persistence time. More importantly, here, it is mediated by cell morphology changes: cell shape and shape variability. Hence we feel these findings are particularly germane to the active matter and biophysics community working on similar problems.

We agree with the referee that our previous section explaining the reentrant dynamics could have been clearer. We have now rewritten this section (*lines: 198-211 of the revised main manuscript*). Also, see *lines: 319-321 of the conclusions*.

Comment 3: In the introduction the authors say "Our experiments unequivocally establish a morphodynamic link, demonstrating that geometric constraints alone can dictate epithelial jamming/unjamming". Where do we see this? If it is so important why this is not commented in the conclusions?

Response 3: We thank the referee for this comment. The morphodynamic link we refer to is captured in Fig. 2, Fig. 3 and Fig. 4. In Fig. 2 we show that the reentrant observed in the dynamics - the structural relaxation time τ_α (Fig. 2b)- is also seen as a reentrant in the cell aspect ratio (Fig. 2d) and the cell shape variability (Fig. 2e). The latter two are related to cell morphology. This is, thus, a morphodynamic link. In Fig. 3, we show that even as the $PDF(AR)$ becomes less skewed, the system becomes more jammed. Again, this is a link between morphology and dynamics, as seen in Ref. [28]. In Fig. 4b, we show that the fast cell population of the system is more elongated than the slow ones and also shows reduced shape variability with respect to the latter. This again is a morphodynamics link.

Following this comment and the referee's previous comment, we have rewritten the conclusions and have attempted to summarize our results better.

Comment 4: The text is not well written:

Response 4: We thank the referee for this comment. We have made many changes to the text to improve the overall readability of the manuscript. If the referee needs specific changes to be made further, we will do so.

Comment 4a: the authors use very complex syntactic structures and very long sentences that make the reading very complicated. For example (but this is not the only one), I cannot understand the following sentences: - Even for the simpler case of rigid self-propelled particles, tuning the activity at constant density results in non-intuitive behavior, such as re-entrant dynamics [34–36], and it is unknown whether similar physics is at play in cell collectives, since, here, the means to tune activity systematically are not available. Synthetic models that embody the key features of confluent epithelia can help bridge the divide between theory/numerical predictions and experiments on living systems. I mean: "similar" to what? which "divide"?

Response 4a: In our revised manuscript, we have avoided using these long sentences and made significant attempts to improve the readability. For the specific sentences mentioned above, the changes made are in *lines: 74-82 of the revised manuscript*.

Comment 4b: the authors use many name and nomenclature without introducing them. For example: confluent tissue, enantiopure and racemic are used without introducing them.

Response 4b: We apologize for not defining the nomenclature clearly. This has been rectified in our revision. The specific terminology pointed out above has been introduced better in *lines: 38, 130 and 131 of the revised manuscript*.

Comment 5: from time to time it seems that the authors are interested more in promoting their experimental setup rather than the results that they have achieved with it. For example authors say ” Recent studies have identified τ_p as a crucial parameter governing active glass physics [33, 34, 36, 44], and our approach allows for systematically tuning it.” But then they do not do it.

Response 5: Earlier simulation studies identified the persistence time of activity as an important parameter governing the physics of dense active matter. There are only a few experimental systems where persistence can be tuned. To the best of our knowledge, ours is the only deformable active matter system where this is possible. We have hence emphasized this salient feature in the manuscript.

We indeed tune the single-cell persistence time, τ_p , by changing the chirality of the cell interior $|\chi_{\text{Cell}}|$. This is summarized in Fig. 1e and in lines: 125-142 of the main manuscript and also in Section III of the Supplement. The persistence time is identified with the time at which the single-cell translational mean-squared displacement transitions from ballistic (slope 2) to diffusive (slope 1) dynamics. The supplementary figure in our previous submission did not have the necessary details. We have now rectified this with a revised figure - **Supplementary Fig. S5A** - that clearly shows how τ_p was determined as the single-cell chirality was tuned. In Fig. 2 b,d, and e and in Fig. 4b, τ_p is the ordinate axis.

Comment 6: The reentrant behavior that seems to be the most important result is indeed very weak... the authors should quantify the errorbars of their measurements. Is it a matter of statistics? How does this scale upon increasing the sample size? is it a finite-size effect? Will it persist for bigger tissues??

Response 6: We thank the referee for this comment. We wish to emphasize that the reentrant behavior that we report is in fact seen for four different ϕ values close to the confluent limit. This was not clearly mentioned in the earlier manuscript but is now (**line 170 of the main manuscript and caption to Fig. 2**). This reentrant in the dynamics (τ_α , Fig. 2b) is also manifest in the cell morphology - aspect ratio (Fig. 2d) and shape variability (Fig. 2e). These quantities are obtained from very different analysis procedures and the fact that they support each other lends strength to our observation. Further, τ_α in our study was picked as the time at which the intermediate scattering function decayed to $1/e$. This is a standard approach (**a new reference has been added [49]**). We did not subscribe to any model of glassy dynamics, as there is no justification yet for doing so, and we did not fit the data; hence, there are no error bars.

Our experimental system, while very versatile because of the many parameters that it allows control over, is also painstakingly assembled by hand. To control τ_p , chiral active ellipsoids within each cell and of the desired handedness are placed manually, one at a time. Assembling the cell collective is a very demanding and time-consuming process. Generating the phase diagram shown in Fig. 2b, where each data point represents a manually assembled experimental system, was a many-month-long endeavor. Larger system sizes would be incredibly more challenging. Having said that, even with a maximum of

only 124 cells on the plate, Fig. 3 shows that our system follows the same scaling as that observed in confluent epithelia. Additionally, the experimental findings in Fig. 4 agree with our simulation results, which were performed for a larger system comprising of 400 cells.

Comment 7: All in all after reading the manuscript multiple time I was left with the feeling that the authors stepped on something indeed interesting but they are not putting their results in a way that the reader can appreciate them.

Response 7: Following the many constructive comments of this referee, we have rewritten parts of the manuscript to convey our results better. This in our view has helped improve the manuscript substantially.

We hope that the referee finds our revised manuscript suitable for publication in Nature Communications.

Comments of Referee 3 along with our responses – NCOMMS-24-01719-T/Arora

The submitted manuscript collective dynamics in a monolayer of synthetic cell-mimics which are driven by vibrating granular particles confined in an elastic boundary. The authors systematically increased the persistence time of self-propulsion and discovered a cell shape-driven, density-independent, reentrant jamming transition. They also observed that cell shape and shape variability are strongly related in the confluent limit and followed the same universal scaling as that observed in confluent epithelia. Furthermore, the fast cells show suppressed shape variability and did not conform to this scaling. These results suggest that geometric constraints alone can dictate epithelial jamming/unjamming.

In my opinion, the submitted manuscript contains novel results and is well written. I recommend its publication after the authors consider the following questions.

Response: We are very pleased to read the very positive assessment of our work by the referee. We thank them for supporting the publication of this study in Nature Communications. Below, we provide a point-by-point response to their comments. Changes to the text are in shown in magenta color.

Specific comments:

Comment 1: The experiments are fairly limited in size: the radius of the imaging area is about 6-7 cell diameter. In such a small system, we should expect strong boundary effects and a spatial gradient over the system. Indeed, cells in the center region of Fig. 1g appear to be more elongated. This spatial inhomogeneity over the whole system, if exists, makes interpretation of the experimental results more complicated. The authors should discuss how strong confinement influences their results and whether it is proper to compare their results with those from an infinite system.

Response 1: We thank the referee for this comment. In the newly added *Section VII of the supplementary text and in Supp. Fig. S12*, we address this concern. At high densities, we had already excluded the layer of cells in contact with the confining wall from the analysis for precisely the reasons pointed out by the referee. To determine if there are any spatial gradients in the cell shape from the confining boundary, we divided the field of view into two regions: an inner radius (R_1) and an annulus (R_2-R_1) shown in Fig. S12(A). These regions were chosen so that they contain approximately the same number of granular cells (38 cells) to ensure a balanced comparison. Fig. S12(B-E) shows the probability density functions (PDFs) of the aspect ratio (AR) for packing fractions $\phi \approx 0.92$ (top panels) and $\phi \approx 0.92$ (bottom panels) and for different τ_p values. For both these values of ϕ and the intervening values, we observed a re-entrant in the dynamics (Fig. 2 b-e of the main manuscript). The statistics for calculating the PDFs are now reduced due to the fewer cells being considered in the analysis. For the image shown in Fig. 1g of the main manuscript, the corresponding PDFs are shown in Fig. S12(B) top panel. We see that the PDFs of the cells in the inner and outer rings almost overlap for all but one value of $\phi = 0.92$ and $\tau_p = 20$ s (Fig. S12(C), top panel). This suggests that the confining boundary does not significantly influence the dynamics (*lines: 225-227 of the revised main manuscript*).

Referee 2 had a similar concern with regard to the system size (comment 6). Our response to this comment is appended below:

Our experimental system, while very versatile because of the many parameters that it allows control over, is also painstakingly assembled by hand. To control τ_p , chiral active ellipsoids within each cell and of the desired handedness are placed manually, one at a time. Assembling the cell collective is a very demanding and time-consuming process. Generating the phase diagram shown in Fig. 2b, where each data point represents a manually assembled experimental system, was a many-month-long endeavor. Larger system sizes would be incredibly more challenging. Having said that, even with a maximum of only 124 cells on the plate, Fig. 3 shows that our system follows the same scaling as that observed in confluent epithelia. Additionally, the experimental findings in Fig. 4 agree with our simulation results, which were performed for a larger system comprising of 400 cells.

Comment 2: The authors used Potts and Vertex models to understand the role of fast cells. I wonder how the parameters of these models were chosen. Are there any experimental supports for the chosen parameters?

Response 2: We thank the referee for this comment. The simulation parameters were chosen such that the relaxation times were comparable to those observed in the experiment. In *Section X of the revised supplement*, we have provided further details. Please also see *lines 281-283 of the revised main manuscript*.

Comment 3: The authors reported a re-entrant jamming transition mediated by cell shape. Is it possible to reproduce such a phenomenon in their numerical models?

Response 3: In our manuscript, we detailed our hypothesis that the observed reen-

trant behavior stems from a balance between cell motility and effective cell-cell adhesion/friction, which, in turn, is influenced by the cells' shape. The essence of this phenomenon is tied to the fundamental relationship between cell shape turgidity and its motility.

To accurately simulate these dynamics in silico would necessitate a more detailed model that integrates cell area elasticity (λ_A) with individual cell motility. Such an exercise, while potentially illuminating, would entail an exhaustive analytical effort that is more appropriately the focus of a dedicated future study.

We thank the referee for their constructive comments, which have helped strengthen our findings and improve the manuscript. We hope that the referee finds our revised manuscript suitable for publication in Nature Communications.

REVIEWERS' COMMENTS

Reviewer #1 (Remarks to the Author):

The authors have revised their paper and answered my comments and questions in a satisfactory manner. I am happy to recommend publication.

Reviewer #2 (Remarks to the Author):

I have appreciated the efforts made by the authors in addressing the issues I have mentioned. I think that the manuscript has significantly improved.

I recommend it for publication.

Reviewer #3 (Remarks to the Author):

The authors have addressed my questions. The manuscript can be published in its current form.